# Insights from a Qualitative Exploration of Adolescents’ Opinions on Sex Education

**DOI:** 10.3390/children11010110

**Published:** 2024-01-16

**Authors:** María Victoria Díaz-Rodríguez, Vicent Bataller i Perelló, José Granero-Molina, Isabel María Fernández-Medina, María Isabel Ventura-Miranda, María del Mar Jiménez-Lasserrotte

**Affiliations:** 1Facultad Ciencias de la Salud, Universidad de Almería, 04120 Almería, Spain; mvtodr@gmail.com (M.V.D.-R.); vicentebatallerp@gmail.com (V.B.i.P.); 2Department of Nursing Physiotherapy and Medicine, University of Almería, 04120 Almería, Spain; isabel_medina@ual.es (I.M.F.-M.); mvm737@ual.es (M.I.V.-M.); mjl095@ual.es (M.d.M.J.-L.); 3Facultad de Ciencias de la Salud, Universidad Autónoma de Chile, Santiago 7500000, Chile

**Keywords:** adolescents, sex education, sexuality, qualitative research, secondary school, family

## Abstract

Background: Adolescence is a crucial time in the development of young people’s identity, and sexuality is a key issue. Comprehensive sex education provides the knowledge and skills to help adolescents protect their sexual and reproductive health and rights. Policies on sex education in secondary schools are highly influential in the development of quality programmes that support comprehensive sex education. The aim of this study was to explore, describe and understand adolescents’ experiences of sex education. Methods: A qualitative study based on Gadamer’s phenomenology was used. Two focus groups and four in-depth interviews were conducted with 12 private school students, followed by inductive data analysis using ATLAS.ti software 9.0. Results: Two main themes were identified in the analysis: (1) sex education is a challenge for secondary schools and (2) student expectations of sex education. Conclusion: It is essential for adolescents to have access to comprehensive sex education that is adapted to the different stages of their development, is provided by specialised teachers, and involves their families in the process.

## 1. Introduction

According to the World Health Organisation (WHO) [1], sexuality is a fundamental aspect of being human, and encompasses sex, gender identities, sexual orientation, eroticism, pleasure, intimacy and reproduction. Sexuality is the way in which people experience and express themselves as sexual beings [2]; psychological, biological, socio-economic, ethical, cultural and religious factors contribute to the achievement of physical and emotional pleasure [1,3]. Individuals experience and express their sexuality through thoughts, desires, behaviours, values, practices or fantasies in a unique way [4,5]. Adolescence is considered a crucial time for establishing one’s identity as young people explore various ways of presenting themselves and “being” in the world [6]. In this search for identity, which they acquire through interactions with friends and family, sexuality is a key aspect [7]. Most adolescents will experience sexual behaviour as a natural and new urge at this age [8]. This experience will be determined by their emotional and social maturity [7,8].

Sexuality and adolescence are two concepts that are intertwined and determined by culture [9,10]. For adolescents, the construction of gender will be of great importance in shaping their sexual identity and sexual behaviours [10]. Identity is one’s self-understanding: the awareness of one’s physical, emotional, mental and sexual self [11]. From a contextual point of view, information from the media and online is crucial in adolescents’ sexual development and the way they experience their sexuality [9]. Social networks have become spaces for young people to express and develop themselves, and cybersex is the first activity through which they can explore their sexuality freely and without prejudice [6]. However, the internet and social networks have also extended access to sexually explicit material such as pornography [12]. Gravningen et al. [13] observed that many adolescents had their first real-world sexual experience after meeting someone online; they used the internet to meet potential sexual partners. This decontextualised exposure to sexual behaviour in adolescence has negative repercussions that can affect their sexual relationships into adulthood, as it can lead to an insensitive approach towards women, thus reinforcing social norms that sustain violence and sexual abuse [14,15]. A lack of understanding regarding sexual needs and sexual health negatively influences adolescents’ future sexual behaviour [16].

The Council of Europe [17] states that “children and adolescents have the right to receive comprehensive, rigorous, scientifically-sound and culturally sensitive sex education, based on existing international standards”. Comprehensive sex education (CSE) provides age-appropriate information, values, attitudes and skills that help protect adolescents’ sexual and reproductive health rights [18]. CSE is fundamental in promoting sexual health and well-being in adolescents, and early interventions will have positive results into adulthood [19]. CSE programmes based on human rights focus on gender equality, the consequences of risky sexual behaviour, and commitment in romantic and sexual relationships, and encourage a culture based on respect, empathy, responsibility and purposeful sexuality [8,20]. The values imposed by society shape the sex life of an individual [21]. The influence of policies on sex education in secondary schools is essential in developing quality programmes that support comprehensive sex education [19]. Spain has a law on sex education in schools, but it is based on a biological and preventive approach, which is easily influenced by political ideology and legislative changes [22]. Different studies highlight the need to incorporate holistic and comprehensive sex education in different autonomous communities in Spain, which depends on the ability to establish sex education policies in the educational sphere [23,24]. In 2021, Organic Law 3/2020 on 29 December (LOMLOE) [25] came into effect. It heavily emphasises gender equality, which can be achieved through joint efforts between education and health systems. It lays the foundations for more innovative education to prevent gender-based violence. However, the current psychological and social aspects of sex education are limited to talks or workshops given by external professionals [26]. School is a prime space for sex education, but it is far from a comprehensive approach, which is even more limited in state schools and universities [27,28].

Previous studies have analysed adolescent sex education from the parents’ perspective [29], the effects of sex education on adolescent sexual behaviour [16] or the relationship between sex education and the reduction in unintended pregnancies among adolescents [30], but little is known about adolescents’ experiences of the sex education they have received. Understanding adolescents’ perspectives on sex education received in Spanish schools allows us to identify their needs and make improvements. The framework developed by Beltran [11] on the development of sexual health and the factors that affect adolescents’ development of gender identity allows us to study adolescent experiences of this phenomenon. The aim of this study was to explore, describe and understand adolescents’ experiences of sex education.

## 2. Materials and Methods

### 2.1. Study Design

A qualitative study was designed based on Gadamer’s hermeneutic phenomenology. For Gadamer [31], to obtain the hermeneutic truth, one must do so through the understanding of the meaning that people assign to their own experiences [32]; meaning is attained through the fusion of the researchers’ and the participants’ horizons. According to Gadamer, understanding a text (transcription) involves dealing with prejudice, culture, tradition and language. The method developed by Fleming and Robb [33] was used in this study. The research question was verified with the methodology used in the study design. Furthermore, the research team’s pre-understanding of the objective was identified.

### 2.2. Participants and Context

The study was carried out in a private secondary school in the province of Almeria. We used convenience sampling to obtain an equal number of students of both sexes. The inclusion criteria were being 15–16 years of age, in the 4th year of compulsory secondary education (ESO) and willing to participate in the study. Exclusion criteria were declining to participate in the study and no consent from the student’s legal guardians. To recruit the sample, one of the researchers arranged an appointment with the school’s management team. All students in the 4th year of ESO were invited to participate. Of the 19 students, 5 did not provide consent from their legal guardians and 2 students did not attend school on the day of the interviews. The final sample consisted of 12 students. The socio-demographic characteristics are shown in Table 1.

### 2.3. Data Collection

The data collection took place in a classroom in the secondary school. The data collection involved two focus groups (FGs) and four in-depth interviews (IDIs) conducted between April and June 2021. Each FG was conducted with students of the same sex with an average duration of 93 min. Subsequent IDIs were conducted with an approximate duration of 62 min. Both FGs and IDIs were carried out by several researchers trained in qualitative research. They followed an interview guide with open-ended questions (Table 2) and recorded their observations in a notebook. Prior to the data collection, the students submitted a signed informed consent form from their legal guardians because this was mandatory for underage participants. All FGs and interviews were audio-recorded. We stopped the data collection when we reached data saturation.

### 2.4. Data Analysis

We transcribed all the audio-recorded FGs and interviews verbatim and used Atlas.-ti software 9.0 for data analysis. The data analysis was conducted following the steps described by Fleming and Robb [33]. Step 1: It was decided whether the research question (Can we study adolescent experiences of sex education?) was relevant. Step 2: The research team’s pre-understanding of the phenomenon was identified. Step 3: We developed an understanding, through dialogues with the participants, in the cases where new themes emerged. Step 4: An in-depth understanding was obtained through a dialogue with the text to identify themes, sub-themes and units of meaning. Step 5: The reliability and rigour of the qualitative data were established in accordance with the Consolidated Criteria for Reporting Qualitative Research [34]. To increase reliability, three members of the research team analysed the data until an agreement was reached. The participants were asked to confirm the transcripts and data analysis.

### 2.5. Ethical Considerations

This study was approved by the Research Ethics Committee of the Department of Nursing, Physiotherapy and Medicine of the University of Almería (code EFM 121/2021), in accordance with the ethical principles of the Declaration of Helsinki. In addition, the school’s head teacher granted permission. The confidentiality and anonymity of the participants was guaranteed in compliance with the Organic Data Protection Act 3/2018 on 5 December on the protection of personal data and the guarantee of digital rights. Permission was requested to record the conversations and an informed consent form was signed.

## 3. Results

### 3.1. Background Characteristics of Participants

All of the participants in the study were 15- or 16-year-old students with Spanish nationality; 50% were male and 50% female. All the participants studied in a private school. Regarding religion, 50% were atheist, 41.7% were Christian and 8.3% were Orthodox Catholic. All the participants were heterosexual.

### 3.2. Themes, Sub-Themes and Units of Meaning

Two main themes emerged from the inductive analysis of the data, which describe the adolescents’ experiences of sex education and what their main expectations of CSE would be (Table 3).

### 3.3. Sex Education: A Challenge for Secondary Schools

We explored adolescents’ experiences of sex education with the question: “What do you think comprehensive sex education should be about? What topics should it cover?” Students’ opinions about the sex education they received in school and their immediate environment would highlight strengths and weaknesses of the current sex education in schools. The participants expressed their concerns, difficulties, and perceived differences in their experiences of sex education.

#### 3.3.1. Students’ Views on the Lack of Sex Education

There are many factors that influence adolescents’ perceptions of the sex education they have received. Sex education in secondary school tends to be biological, focusing only on the anatomy and functioning of the male and female reproductive system. Most of the sessions are focused on issues of reproductive processes, in particular, the menstrual cycle of women.

“*In biology, what we have learned, the reproductive system and how reproduction works*”.(MFG-4)

“*I would have liked to receive more information, not only focused on biology*”.(IDI-4)

The sex education received by adolescents is often aimed at the prevention of sexually transmitted infections, with a particular focus on condoms as a barrier method. The students consider this topic important because they are aware that using condoms reduces the chances of pregnancy and STIs. However, the adolescents felt that they did not go into enough depth and that they were taught the same thing every year, regardless of their age. They would have liked to have learned about the emotional, psychological or social factors that may affect them. The lack of more inclusive and comprehensive sex education left students with unanswered questions and fears.

“*Whenever they have talked to us about contraceptive methods, it is always really basic, when really, there are loads of things because, for example, this summer I found out when I went to my aunt’s house that there were loads of methods that I didn’t even know existed, like the female condom. It’s always the same too. I think that because they are not entirely comfortable, they talk about what we already know because we already know it*”. (FFG-3)

“*I think all that is good, to avoid pregnancy; but on the other hand, you have to work on the social part, because suppose you get pregnant and then you hear comments that it is reckless or something like that*”.(IDI-3)

It should be noted that most of the participants in the study did not know what sex education entailed. Many of the adolescents agreed that the sex education they had received at secondary school had been very poor or non-existent. They stated that the little knowledge they had acquired had come from their parents or from what they had studied in biology classes on reproduction.

“*Personally, I studied in the United States, and they gave me classes on prevention and things that I see as normal, but not here in Spain*”.(FFG-1)

“*Well, in 2nd of ESO, we were taught the reproductive system. That’s all I know. Neither my parents nor the school taught me anything else*”.(IDI-3)

It is interesting to note that most of them could not rely on their families as a source of information and had nobody to respond to their questions. Therefore, many of them turned to the internet to look for information, resulting in self-directed learning.

“*For example, I have an older sister and neither she nor my mother have ever explained anything to me, neither how to put in a tampon nor how long a sanitary towel lasts. I had to look for information myself on the internet, but I always looked on more than one website*”. (FFG-4)

“*On social networks, if I know that someone around me, like a friend, knows about it, I ask them, but on the internet*”.(IDI-2)

#### 3.3.2. Main Problems and Difficulties in Secondary Schools

The participants described that their main difficulty was the barriers they encountered when talking about certain topics with their teachers. They observed that the teacher was not comfortable or trained to respond to their concerns on issues related to sex education. As one of the participants stated, they avoid questions or respond in an evasive way.

“*I think that there are also times when teachers don’t feel confident or don’t see it as the most normal thing in the world. They talk to you about contraception or the morning-after pill, but they don’t feel comfortable talking to us about it*”.(FFG-3)

It is worth noting that the majority of the participants feel that their teachers always repeat the same topics; they stick to the textbook and cover the same subjects that they have been learning since sixth grade. The adolescents feel that as they get older, they have a need to further their awareness in order to understand their experiences of sexual behaviour or identity. They feel that CSE could cover much more. As one participant stated, it is not enough to focus on one didactic unit in the biology textbook.

“*Societal opinions, hygiene and above all depilation. If I really don’t feel like shaving, how will the other person react? The issue of sexual orientation, if I decided to kiss a girl now, what would people think?*”(IDI-3)

“*And besides, whenever they talk to us about this, they always tell us the same things, the same thing in the second year of ESO as in the first year of ESO. I think there is a big difference, we are more mature now and can talk about more specific issues*”.(FFG-2)

Secondary schools face many difficulties in providing sex education. As one participant described, too much importance is given to core subjects, which, although important, do not prepare you for everyday life. Furthermore, all of the adolescents agreed that teachers have to contend with laughter and embarrassment in the classroom. Most of the participants admitted that they lacked maturity when it came to the subject of reproduction.

“*Laughter, especially among the boys, and it is understandable because we are young and more immature*”.(IDI-3)

“*A few laughs. We women normally mature earlier than men so of course, when we saw the external part of the reproductive system in 2nd of ESO, there was lots of giggling. The person who had to read couldn’t concentrate. As they gave us the books in September, there was already someone who was going to find that topic and try to be funny*”.(IDI-1)

#### 3.3.3. Gender-Related Needs

The participants described differences in their sex education needs. The teenage girls showed more interest in issues related to their sexuality and how to experience it. They also had more concerns about their bodies, highlighting feelings of shame and self-esteem. Some participants were concerned about how to behave when undressing for the first time in front of a boy or even what society would think if they saw them kissing a person of the same sex. Adolescence is a time of many physical and psychological changes that bring about feelings of self-consciousness and embarrassment among teenage girls, as several of the participants expressed.

“*Well, maybe the topic of hygiene, how I should shave or how it is accepted that I have to shave or... the moment when I have to see myself completely naked in front of another person, because of the stereotypes nowadays*”.(IDI-1)

“*Everything that has to do with sexuality is important to me. To know how I might feel in that moment, what I can expect and how to act*”.(IDI-3)

Most of the teenage girls said that it was important for them to further their knowledge about how to live their sexuality or how to enjoy it. As the interview progressed, themes such as female masturbation emerged. Several of the participants pointed out that they did not feel comfortable talking about this topic; it was unknown and taboo. They also felt that while it was natural for boys, it was unthinkable for them.

“*For example, 3 years ago I found out that you can also touch yourself and I didn’t know that*”.(FFG-2)

“*It was unthinkable to talk about it with your friends, but the boys were already talking about it in the first year of ESO*”.(FFG-3)

On the contrary, the teenage boys’ greatest concern was related to sexual intercourse. They were interested in knowing how they should do it in order to feel pleasure and at the same time satisfy their partner, especially the first time. Some of the participants highlighted their concern about female pleasure.

“*My main concern is that when the time comes, I don’t satisfy the other person, that I disappoint her and that she doesn’t feel good because I did it wrong or something*”.(MFG-2)

“*For example, they could also teach us how to do it well and how to make the other person and you enjoy it*”.(MFG-1)

### 3.4. Student Expectations of Sex Education

This theme reflects the adolescents’ expectations of the sex education they would like to receive. Most of the participants felt ready to receive CSE, to be treated as adults and to establish a foundation for joint approach to education between family and school.

#### 3.4.1. Comprehensive Sex Education

The participants believe that CSE is not merely knowledge about reproduction and the health risks associated with sexuality. These adolescents want to learn about issues related to gender stereotypes, sexual relationships, sexual orientation and pleasure. They stated that LGBTIQ+ people are often completely excluded by society. Therefore, they felt that understanding what sexual orientation or gender identity was would help them to break down the existing stereotypes. They also stated that most of the time, they searched for their answers on the internet.

“*Normalise things, teach us the basics, stereotypes and understanding our sexuality. I think that would make society better*”.(IDI-3)

“*We live in a society in which we are gradually improving the issue of sexual rights and sexual orientation, but there are still people who don’t accept it and make memes*”.(IDI-2)

Another important issue for the participants is learning about more emotional, cognitive, social and ethical aspects of sexuality. As several of the adolescents pointed out, they would like to receive education on skills, attitudes and values in order to feel more empowered. They want to develop respectful sexual relationships that make them feel good and not have a distorted image, as shown on social media. It is a complex age that brings many physical and psychological changes. In this context, they feel that they do not conform to social stereotypes and are concerned about what society expects of their sexual behaviour.

“*I believe that our society, when it comes to sexual relations, is based on trial and error because we are never informed. If you make a mistake, then you are wrong for life. How do I know if I’m doing it right?... I’m afraid of making mistakes that can’t be fixed... I don’t feel confident*”.(FFG-4)

“*Self-esteem, I am very affected by people’s comments. I pay a lot of attention to others so as not to be different to the norm... what if they don’t like me?*”(IDI-3)

#### 3.4.2. Uncensored Sex Education

The participants were very clear about their wishes and expectations. Adolescent boys and girls want teachers to stop treating them like children and start treating them like adults. They want to be able to have their questions and fears answered and want to know more about their sexuality. Yet, they feel that they are being deprived of this right.

“*I think that the subject has always been very taboo, but that could stop and that information could be provided because it is something that’s normal and we want to know everything. Because, of course, in the end, we are the ones who end up being harmed by not talking about all these things*”.(MFG-1)

“*I think that school should also help you with the fear of the first time or the fear of having an abortion, because there are many fears, like the fear of getting pregnant*”.(FFG-2)

It is important to highlight that the participants expressed a desire to be able to talk to teachers about virginity, sexual relationships and pleasure. They described how they feel afraid and embarrassed to discuss certain topics. They feel uninformed about relevant topics, which leads them to look for answers on the internet. They also felt judged at times for certain sexual behaviours. One participant said that she felt judged and even stigmatised because of what people would think of her if she masturbated.

“*When it comes to masturbation, it’s always much more open with the boys whereas if a girl says, well, I have done such and such they say, seriously?*”(FFG-3)

“*They should educate us about what it is, if for example virginity is something that all women have or not, because I have heard that there are times when a woman does not bleed, and these are things that no one tells us. We either find out through social networks or what we talk about among ourselves*”.(FFG-4)

#### 3.4.3. Joint Approach to Education: Family and School

The participants were of the opinion that they should be educated about sexuality. They have the right to be able to further develop their emotions, make decisions and establish healthy relationships. However, they did not all agree on who should teach them these values and information. Some of the adolescents believed that school is the right place to receive this education, stating that teachers could take a more formal or scientific approach. They would also feel more comfortable to ask questions, which they would not know how to ask at home.

“*School is the ideal place because they know us, you can discuss issues and answer questions the students have in common as a class. I would find it more uncomfortable with my family. I don’t see myself talking about this topic with my parents. I wouldn’t even know how to start the conversation with my family. With people my age yes, but not with older people*”.(IDI-3)

However, other participants believed that there should be a joint approach to education involving the school and the family. They stated that in a more informal environment, such as the home, they feel more comfortable to talk about values or emotions, whereas at school, it can be more difficult for them to ask certain questions in front of their classmates. However, when it came to talking about other topics, they felt that it was better to deal with them in class because they had questions that the rest of their classmates would also like answered. Therefore, they believe that the ideal scenario would be a joint approach to CSE as it would provide them with different sources of information.

“*I think that if there is no communication between the school and the family, they won’t be able to teach you something well, so it’s better to involve both*”.(FFG-3)

“*At home I feel more confident to talk about certain issues. I usually ask my mother because she doesn’t mind talking about these things*”.(FFG-1)

## 4. Discussion

This study aimed to explore, describe and understand adolescents’ experiences of sex education. The framework developed by Beltran [11] allowed us to understand the participants’ experiences of the sex education they have received, as well as to identify the needs and concerns they have at this stage of their development. The participants considered the sex education provided by their secondary school to be insufficient. This is in line with other studies that showed the need to restructure the school curriculum to include more aspects relating to sexuality and sex education in the syllabus [35,36]. Most secondary schools focus exclusively on teaching anatomy, reproduction and prevention of sexually transmitted diseases [37]. The adolescents in this study value education about the anatomy and physiology of the reproductive system as an essential element in their education [38]. However, they pointed out that it was necessary to address not only the physical changes but also the psychological, emotional and social changes during puberty [9]. Awareness of the changes that occur during puberty increases confidence and is key to developing appropriate sexual health behaviours [39]. Another topic taught at secondary school is identifying the most common sexually transmitted diseases. The participants named the condom as the main barrier method [20]. Although evidence shows that there is a high percentage of adolescents who do not use barrier methods when they have sex [5,40], secondary schools continue to provide very little information [41]. Adolescents have limited knowledge about the consequences, use and types of barrier methods, which leads them to search on the internet [42]. Seeking information on the internet and using interactive sexual and reproductive health services gives adolescents a greater perception of confidentiality and privacy [43]. The participants expressed concerns, fears and needs related to their sexual health and behaviours that were not being met by teachers [44]. The needs and concerns expressed by the participants regarding sex education differed according to gender. In line with López et al. [10], they pointed out that despite these differences, the ideal situation would be for CSE to facilitate adolescents to live their sexuality as they wish. Sex education should allow them to talk about emotions and feelings, while at the same time providing the knowledge and skills they need to have satisfying and safe relationships and to be responsible for their sexual health [44]. In addition, there is a need for comprehensive training for teachers so that they can deliver effective CSE [36].

The study participants believe that CSE should include information related to sexual orientation, gender identity, prevention of gender-based violence and other aspects that help to break down existing stereotypes [45]. With regard to sexual orientation and identity, the findings of this study suggest that the adolescents’ lack of knowledge in this area leads to conflict. Furthermore, most resort to seeking information on social networks [46]. McBride [47] points out that the education provided in schools is based on the gender binary, focusing on the link between anatomy and gender identity, as well as the binary division of male and female. In recent years, attempts have been made to integrate perspectives on gender and LGBTQI+ rights in schools in order to prevent homophobia, bullying and gender-based violence, but it is still insufficient [48,49]. On the other hand, sexuality has been strongly influenced by feminist movements [50]; basing sex education on these ideals is a transformative goal for achieving significant change [49]. Adolescents want and expect to develop respectful sexual relationships. They point to the need for CSE that strives to deconstruct societal stereotypes that distort reality and undermine their self-esteem [51]. Over the years, educational institutions have produced an explicit curriculum in which history was studied from the point of view of men, a hidden curriculum of sexist behaviour that omits the teaching of the contributions that women have made to society [52]. CSE that promotes human rights and gender equality and improves adolescent sexual and reproductive health must be pursued [53]. This study highlights the need for a joint approach to CSE between the secondary school and the family, including improved teacher training [36] and giving a prominent role to the family [35]. The family must provide sound sex education at home to prepare adolescents for satisfying relationships and healthy sexual behaviour [37,46].

### 4.1. Limitations

All of the participants study in the same private secondary school in the south of Spain. Participants from other secondary schools or countries could change the results. The study sample is homogeneous in terms of age, social class, culture and educational background. In order to compare the results, studies could be carried out with a more heterogeneous sample. Another limitation is the small study sample due to the number of students at the school and the number of adolescents who attended school on the day of data collection. Using a measure of social desirability bias together with other strategies could improve the interpretation of the findings.

### 4.2. Recommendations for Future Research

Future research could explore the development of teacher training programmes with the aim of adapting emotional and sex education to the students’ level of maturity throughout the different stages of education. The complex nature of addressing sex education means that teachers need to be adequately trained to facilitate interactions with students. Teachers should therefore receive training that enhances their knowledge and skills so that they are able to teach sex education. The development of innovative sex education programmes requires appealing educational and pedagogical strategies.

## 5. Conclusions

Exploring adolescents’ experiences of school-based sex education provides an insight into their needs and the challenges they face. Sex education in secondary schools is poor and focuses on biological concepts and the prevention of sexually transmitted diseases. Adolescents feel prepared to address the social, psychological, emotional and ethical fators related to sexuality. They are calling for education that includes information related to sexual orientation, gender identity and the prevention of gender-based violence so that they can develop respectful sexual relationships. For CSE to be comprehensive, it requires a gender- and diversity-focused approach that facilitates the identification of gender inequality and how to deal with it, as well as awareness of one’s own body. Schools should convey clear messages promoting gender equality, non-stereotypical gender roles, consensual sexual relationships and respect from the earliest stages of education. Designing curricula that incorporate CSE would help to tackle problems such as gender-based violence, discrimination against women, sexual harassment, cyber-bullying, bullying and social rejection of the LGBTI community. CSE should be improved and adapted to the different developmental stages, using specialised teachers and involving families in the process. The results of this study can be used to illustrate the importance of developing regulatory frameworks that support comprehensive sex education and encourage this change in education.

## Figures and Tables

**Table 1 children-11-00110-t001:** The socio-demographic characteristics of the participants (N = 12).

Participant	Age	Sex	Sexual Orientation	Gender Identity	Profession of Legal Guardians	Type of Secondary School	Religion
FFG-1	16	Female	Heterosexual	Woman	Justice civil servants	Private	Christian
FFG-2	15	Female	Heterosexual	Woman	Nurse and self-employed	Private	Atheist
FFG-3	15	Female	Heterosexual	Woman	Elementary school teacher and economics teacher	Private	Atheist
FFG-4	15	Female	Heterosexual	Woman	Self-employed	Private	Christian
MFG-1	15	Male	Heterosexual	Man	Diplomat	Private	Orthodox Catholic
MFG-2	15	Male	Heterosexual	Man	Firefighter and civil servant	Private	Atheist
MFG-3	15	Male	Heterosexual	Man	I.T. engineer	Private	Atheist
MFG-4	15	Male	Heterosexual	Man	Prison civil servants	Private	Atheist
IDI-1	15	Female	Heterosexual	Woman	Builder and social worker	Private	Christian
IDI-2	15	Male	Heterosexual	Man	Nurse and administrator	Private	Christian
IDI-3	15	Female	Heterosexual	Woman	Doctor and hairdresser	Private	Christian
IDI-4	15	Male	Heterosexual	Man	Self-employed	Private	Atheist

FFG: female focus group; MFG: male focus group; IDI: in-depth interview.

**Table 2 children-11-00110-t002:** Interview guide.

Stage	Subject	Content/Possible Questions
Introduction	My intention	To learn about adolescents’ experiences of sex education to understand their concerns and needs regarding CSE.
	Ethical issues	Inform about the voluntary nature of participation, registration, consent, data confidentiality and the possibility to withdraw from the study at any time.
Beginning	Introductory question	At any point in your education, have you received information related to sex education, sexuality or human sexuality? What was this information like?
Development	Conversation guide	What do you think comprehensive sex education should be about? What topics should it cover? When you want to get information about sexuality, where do you look? What topics about sex education do you usually talk about? What issues related to your sexuality are you most concerned about? How could these be solved? From whom would you like to receive sex education, and why (family, school, etc.)? Imagine you have a subject called Comprehensive Sex Education. What would you want to study in it?
Closing	Final question	Is there anything else you would like to add?
	Appreciation	Thank them for their participation, remind them that their interview will be of great use, and place ourselves at their disposition.

**Table 3 children-11-00110-t003:** Themes, sub-themes and units of meaning.

Theme	Sub-Theme	Units of Meaning
Theme 1: Sex education is a challenge for secondary schools	Students’ views on the lack of sex education	Anatomy, contraception, lack of sex education, family, internet, STIs, menstruation, same old same old
	Main problems and difficulties in secondary schools	Embarrassment, taboo, textbook, teacher indifference, maturity
	Gender-related needs	Sexual relationships, differences, health, masturbation, self-esteem
Theme 2: Student expectations of sex education	Comprehensive sex education	Autonomous learning, self-esteem, change in mentality, beginning to know, relationships, knowing how to act
	Uncensored education	Fears, sexual orientation, sexual relationships, sexual health, society, taboos
	The family and the school: a joint approach to education	Family participation in secondary school education, society, taboos, ethics

## Data Availability

The data presented in this study are available on request from the corresponding author. The data are not publicly available due to confidentiality purposes.

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
