# Peer review of "Insights from a Qualitative Exploration of Adolescents’ Opinions on Sex Education"

_children, 2024, doi:10.3390/children11010110_

Round 1
Reviewer 1 Report
Comments and Suggestions for Authors
Review for Children #2809459
The present manuscript, titled “Adolescent Experiences of Sex Education: A Qualitative Study,” seeks to describe and understand the experiences of 12 adolescents with sex education. I appreciate the insights provided by the current study into this important topic as there is still a dearth of research on sexuality education among adolescents. I enjoyed reading this manuscript, and believe it will be of interest to the readership of Children. However, I have some suggestions for the authors to remove the manuscript, noted below:
1. First, considering the study’s focus on the perspectives of only twelve participants, I might advise the authors to revise the title to avoid any unintended implications of a broader, national evaluation. One example along these lines might be something like “Insights from a Qualitative Exploration of Adolescents’ Opinions on Sex Education” – a minor adjustment, but one that I believe could enhance the clarity of your paper’s focus and help readers accurately interpret the scope of your research.
2. Page 2: “Gravningen et al [13] noted that many adolescents had their first sexual experience online” – what does this mean? An extra sentence here may be helpful to provide context.
3. Along similar lines, I think the paragraph about online sexual experiences and learning is a good setup for your study finding about participants turning to the Internet to fill the gaps in sexuality education left by school curricula, but I think a better case can be built in this paragraph for why adolescents might do this. There is a body of literature suggesting the effectiveness of online approaches in improving sexual knowledge, attitudes, and behaviors among adolescents. The Internet also provides a space for adolescents to receive sensitive/taboo information in a comfortable and private way, avoiding potentially awkward conversations with friends or family members. If not in the Introduction, then perhaps this would be suitable in the Discussion.
4. It is not until the Methods that we receive any indication that this research took place in Spain. Would it be possible to set up in the Introduction why this research is important to be conducted there specifically, or details on the state of sex education in Spain compared to other places in the world?
5. Also, in the Introduction it might be helpful to distinguish any established differences between private and public school sex education, considering that all of these participants experienced private school sex education. I would also include in the Abstract that the sample came from a private school as that is a unique characteristic of the sample that may limit generalizability.
6. Please provide information about how you got from all 4th year of ESO students being invited to participate (how many is this) to only 12 participants being included in the final sample. How many were excluded due to lack of parental consent vs their own decision not to participate? Did the researchers consider a waiver of parental consent or believe that this may have helped with recruitment?
7. In Table 1, one of the “male” gender labels says “hombre” – unsure if this was intentional or a mistake.
8. In Table 1, what does the “Sexual identity/gender” label mean? I am assuming it is just their gender identity; if so, please reword.
9. In-depth interviews are labeled “I” in the table and “IDI” in text. I would suggest being consistent with one term (I recommend “IDI”) to avoid any confusion.
10. Some of these findings are very rich and prompt further discussion and implications. I would advise that the authors provide more interpretation and recommendations for future research and intervention in the Discussion section. For example, in Results section 3.1.2, you discuss the teachers not being trained or comfortable in the material that they are teaching. What would you recommend, based on this finding? For example, future research could explore the development and effectiveness of specialized teacher training programs focused on enhancing educators' comfort and competency in addressing sensitive sexuality-related topics. In terms of policy recommendations, perhaps there could be incentive programs for teachers participating in professional development or continuing education related to sex education. These are just some ideas, but I feel as though much more could be said in the Discussion section to expand on the specifics of your findings, if the world limit allows.
11. A big critique is the lack of Limitations. I strongly suggest including more limitations in addition to the one sentence you have about the limited generalizability of the sample. Was there a risk of social desirability bias? What about the potential bias in participant selection, with certain participants excluded based on the methods getting to the 12 participants?
Author Response
Dear reviewer, I am attaching a document file with the responses to the comments.
Thank you very much,

Reviewer 2 Report
Comments and Suggestions for Authors
Abstract
Line 22- remove the “s” on conclusions.
Line 25- key words: change comprehensive sex education to sex education.
1. Introduction
Line 67- Change this sentence “. One's sex life is shaped by the values imposed by society [21]” to “The values impose by society shape the sex life of an individual [21].”
2. Materials and Methods
2.2. Participants and Context
Line 92-93: Change this sentence- “Convenience sampling was carried out so that there would be an equal number of students of both sexes.” To “We used convenience sampling to have equal number of students of both sexes.”
Line 93-95 to read as “The inclusion criteria were 15—16 years of age, in the 4th year of compulsory secondary education (ESO) and willing to participate in the study.”
Line 95-96: Exclusion criteria were declining to participate and no consent from students’ legal guardians.”
Line 98 to 101: Move Table 1 to the first section of Results.
Add: All FGs and interviews were audio-recorded.
2.3. Data Collection
Line 103-104- to read as “The data collection took place in a classroom in the secondary school.
Line 109- 112: Change to “Prior to data collection, students received an informed consent form from their legal guardians because the authorization was mandatory for underage participants.”
Line 112: Change to “We stopped data collection when we reached data saturation.”
2.4 Data Analysis
Then Start the section with’ “We transcribed all the audio-recorded FGs and interviews verbatim and used Atlas-ti software for data analysis.”
Line 124: The use of triangulation if not correctly used here and nor required here because the authors used only two sets of data FGs and interviews. Therefore, I suggest delete lines 124 to 126.
Also remove “Lastly.”
3. Results
3.1. Background Characteristics of Participants- Start with Table 1 and provide a brief written description of the table.
3.2. Themes, Sub-Themes and Units of meaning- Table 3
Move lines 138-140 after the table 3.
3.3. Sex education: A Challenge for Secondary Schools
Lines 143-146: to read as= “We explored adolescent experiences of sex education with the question on “What do you think comprehensive sex education should be about? What topics should it cover?” Students’ opinions about sex education received in school and their immediate environment would highlight the strengths and weaknesses in the present sex education in schools. The participants expressed their concerns, difficulties, and perceived differences in their experiences of sex education.”
3.1.1 to be 3.3.1 Students' Views on the Lack of Sex Education
Lines 162-163: to read as- “The lack of a more inclusive and comprehensive sex education left students with unanswered questions and fears.”
3.1.2 to read as 3.3.2. Main Problems and Difficulties in Secondary Schools
3.1.3 to become 3.3.3. Gender-Related Needs
Lines 248: Start as- “On the contrary, teenage boys greatest concern was related to …….
3.2. to be 3.4. Student Expectations of Sex Education
Line 261 3.2. to be 3.4.1 Comprehensive Sex Education
3.2.2 to become 3.4.2 Uncensored Sex Education
3.2.3 to become 3.4.3 Joint Approach to Education: Family and School
4. Discussion
Line 338: remove the word “has” from “The framework developed by Beltran [11] has allowed us to ….
Line 356: The study’s participants—” delete “study’s.”
Generally, have a space between the quotes/excerpts.
Comments on the Quality of English Language
Okay.
Author Response

(The authors gave the same response as above.)

Reviewer 3 Report
Comments and Suggestions for Authors
The paper titled "Adolescent Experiences of Sex Education: A Phenomenological Exploration" presents a qualitative study that aims to explore, describe, and understand the experiences of adolescents living in the South of Spain regarding sex education.
The strength of the paper lies in its clear articulation of the research objective and the relevance of the study. The emphasis on the importance of comprehensive sex education for adolescents, especially in the context of their sexual and reproductive health and rights, is well established. The utilization of Gadamer's phenomenology for the qualitative study design adds depth to the research methodology.
However, I think that there are areas for improvement, in order to enhance the paper's relevance to a broader readership, thereby contributing to the ongoing discourse on adolescent sex education.
First, the paper lacks clarity regarding why the audience of the journal should be interested in a specific study conducted in the south of Spain. It is crucial to elaborate on the unique aspects or characteristics of the case study that make it relevant and potentially applicable to a broader audience. Providing insights into the generalizability of the findings or how they contribute to existing knowledge on sex education would enhance the significance of the study. Accordingly, the paper would benefit from a dedicated section addressing the cultural context. Including information on existing similar initiatives, the political landscape surrounding sexual education in schools, and any notable regional nuances would add depth to the study. This contextual information is essential for readers to understand the external factors influencing the study and to evaluate the transferability of the findings to different cultural settings. In the methodological part, discussion of research ethics contains no reference to ethical approval by the relevant institution’s research ethics committee (or equivalent). The conclusion part of the paper mentions the role of feminism and LGBT+ studies to inform sexual education initiatives, but it lacks specificity. Authors should elaborate on how these theoretical frameworks can practically inform sex education, detailing the specific topics, methods, or approaches that could be influenced. Providing concrete examples or recommendations would strengthen the conclusion and offer actionable insights for educators, policymakers, and researchers. In the final part of the paper I suggest to integrate specific aspects related to gender and sexual minority issues. Topics such as gender-based violence, STDs prevention, and diverse gender identities (e.g., bisexuality, agenderism, non-binarity) are critical components of comprehensive sex education. Addressing these issues would make the study more inclusive and relevant to the diverse needs of adolescents, ensuring that the proposed sex education initiatives cater to a wide spectrum of identities and experiences.
Overall, I recommend accepting the paper with minor revisions. The addition of suggested information would strengthen the work, providing a more comprehensive understanding of the study's implications and contributing to the existing literature on adolescent sex education.
Author Response

(The authors gave the same response as above.)

Round 2
Reviewer 1 Report
Comments and Suggestions for Authors
I appreciate that the authors addressed the reviewers' comments so thoroughly. I believe the paper is suitable for publication in its current form.